# Blood Biomarkers of Response to Immune Checkpoint Inhibitors in Non-Small Cell Lung Cancer

**DOI:** 10.3390/jcm11113245

**Published:** 2022-06-06

**Authors:** Yolanda Lage Alfranca, María Eugenia Olmedo García, Ana Gómez Rueda, Pablo Álvarez Ballesteros, Diana Rosero Rodríguez, Marisa Torres Velasco

**Affiliations:** Department of Medical Oncology, Hospital Universitario Ramón y Cajal, 28034 Madrid, Spain; maruolmedogarcia@hotmail.com (M.E.O.G.); anagrueda@gmail.com (A.G.R.); pabloteros@gmail.com (P.Á.B.); chabis25@gmail.com (D.R.R.); marisa7tv@hotmail.com (M.T.V.)

**Keywords:** non-small cell lung cancer, immunotherapy, blood biomarker

## Abstract

Immune checkpoint inhibitors (ICIs) have revolutionized the treatment landscape of non-small cell lung cancer (NSCLC), either used in monotherapy or in combination with chemotherapy. While some patients achieve durable responses, some will not get benefit from this treatment. Early identification of non- responder patients could avoid unnecessary treatment, potentially serious immune-related adverse events and reduce treatment costs. PD-L1 expression using immunohistochemistry is the only approved biomarker for the selection of patients that can benefit from immunotherapy. However, application of PD-L1 as a biomarker of treatment efficacy shows many deficiencies probably due to the complexity of the tumor microenvironment and the technical limitations of the samples. Thus, there is an urgent need to find other biomarkers, ideally blood biomarkers to help us to identify different subgroups of patients in a minimal invasive way. In this review, we summarize the emerging blood-based markers that could help to predict the response to ICIs in NSCLC.

## 1. Introduction

Lung cancer is the second most common cancer and the leading cause of cancer-related death worldwide [1]. Until recently, platin-based chemotherapy has been the standard treatment for advanced patients with non-small cell lung cancer (NSCLC) without driver mutations [2,3,4]. However, immunotherapy, a wealth of new approaches involving the stimulation of the immune system, has revolutionized the treatment of lung cancer [5].

So far, immune checkpoint inhibition with monoclonal antibodies targeting programmed cell death protein 1 (PD-1) or its ligand (PD-L1), such as nivolumab [6,7], pembrolizumab [8,9,10,11], cemiplimab [12], and atezolizumab [13] have demonstrated improvement in overall survival (OS) and progression-free survival (PFS), both as monotherapy regimen and in combination with platin-based chemotherapy in patients with advanced disease. Furthermore, durvalumab was approved in patients with unresectable stage III NSCLC after concomitant chemoradiotherapy [14,15].

On the other hand, the IMpower010 trial showed a disease-free survival benefit with atezolizumab versus best supportive care after adjuvant chemotherapy in patients with completely resected stage II-IIIA NSCLC (HR 0·81 (0·67–0·99; *p* = 0·040)) which offers a promising new treatment option for patients with resected early stage NSCLC [16].

In the neoadjuvant setting, the Checkmate 816 phase III trial also showed that the combination of nivolumab plus chemotherapy improved the pathologic response rate, in comparison to chemotherapy alone (median residual viable tumor cells in the primary tumor bed were 10% vs. 74% respectively) [17].

Currently, the expression level of PD-L1 measured by immunohistochemistry (IHC) in NSCLC tumor samples is the sole biomarker approved by regulatory agencies to guide the decision treatment [18]. However, even though immune checkpoint inhibitors (ICIs) have documented long-lasting responses in some cases, their benefit in NSCLC is limited to a subset of patients. Although these treatments are generally well tolerated, a number of severe immune-related adverse events have been documented, highlighting the relevance of identifying properly the patients who can benefit from ICIs [19,20,21,22]. Furthermore, several studies have pointed out the lack of strength of this biomarker when predicting the efficacy of ICIs. On the one hand, it is not uncommon to identify subsets of PD-L1 negative patients with long-lasting responses after receiving ICIs as single therapy. On the other hand, some patients with very high expressions of PD-L1 do not respond to ICIs, suggesting that there are more factors involved in the process, including tumor heterogenicity [23] or site of biopsy [24]. It has been documented that the expression of PD-L1 can be lower in the primary tumor than in samples from metastatic disease. In addition, dynamic changes in the expression of PD-L1 over time and related to the different antitumoral treatments have been also suggested. Therefore, in the last few years, different biomarkers are being analyzed. Tumor mutational burden (TMB), neoantigen load, tumor-infiltrating lymphocytes, and immune-regulatory mRNA expression signatures are some of the different areas being explored nowadays. However, obtaining an adequate tumor sample in advanced lung cancer patients can be an important limitation, and make serum markers a very attractive research area.

Due to the limitations exposed, a remarkable effort is being applied in order to identify the potential role of blood biomarkers used in routine clinical practice as predictors of response to immunotherapy, which may imply an amelioration in cost effectiveness and also, given the facility of extraction, allow a dynamic verification or the treatment’s response (Figure 1).

In this review, we describe the potential predictive and/or prognostic role of blood biomarkers in the setting of advanced NSCLC patients treated with ICIs (Table 1 and Table 2).

## 2. Peripheral Blood Cell Count: Lymphocytes, Neutrophils, Eosinophils, Monocytes, and Platelets Count

Systemic inflammation has been linked with cancer development, cancer cachexia, and overall poor outcomes. The presence of proinflammatory cytokines such as interleukin 6 (IL-6), and tumor necrosis factor α (TNF-α) in the tumor microenvironment induces myelopoiesis, erythropoiesis and causes peripheral changes in the white blood cell count such as neutrophilia; these neutrophilia can suppress the cytolytic activity of lymphocytes, activated T cells, and natural killer cells [25].

Routine blood parameters have been also investigated as potential inflammatory biomarkers in patients with several cancer types, including NSCLC. For instance, it has also been described that high neutrophil but low lymphocyte infiltration in the tumor microenvironment could promote angiogenesis and inhibit cell apoptosis, which ultimately would favor tumoral growth and therefore would be a poor prognosis data [26]. Based on that, the neutrophil to lymphocyte ratio (NLR) has been studied as a potential biomarker.

A retrospective study of 157 patients treated with pembrolizumab and nivolumab showed that baseline neutrophilia, defined as an absolute neutrophil count (ANC) of 7.5 × 109/L or higher, was associated with a worse prognosis. The overall survival (OS) at 12 months was 34.9% (95% CI, 19.0–59.0) for patients with neutrophilia compared to 42.9% (95% CI, 33–55.6) for patients with lower baseline ANC (*p* = 0.01). After adjusting for age, sex, ECOG performance status, and the number of lines of chemotherapy, a high baseline absolute monocyte count (AMC) was also significantly associated with an increased risk of death (HR, 1.71; 95% CI, 1.06–2.75; *p* = 0.028) and progression (HR, 1.50; 95% CI, 1.06–2.12; *p* < 0.03). However, the baseline absolute eosinophil count (AEC) and platelet counts were not associated with tumor response and/or treatment outcomes. Finally, patients with a baseline NLR of 5.9 or higher showed inferior 1 year OS (31.9% (95% CI, 19.3–52.7) in comparison to those patients with lower baseline NLR (47.3% (95% CI, 34.6–61.6) (*p* = 0.004)). Similar differences were found when analyzing progression-free survival (PFS), which showed 6 month PFS of 14.4% in patients with higher baseline NLR compared to 31.1% in patients with lower baseline NLR [27]. However, different data have been reported in another retrospective study of 55 metastatic cancer patients (being the most represented tumor type NSCLC (33%)). Although baseline NLR levels did not have a significant impact on PFS, the increased NLR after two cycles of anti-PD-1/PD-L1 therapy had a negative effect on PFS in the univariate analysis (HR 1.14, 95% CI 1.06–1.23, *p* = 0.004). These results suggest that not only baseline levels but dynamic changes in NLR could be relevant in predicting outcomes [28].

A question that still remains to be solved is the optimal cut-off for NLR. A meta-analysis of 3656 advanced NSCLC patients reported NLR > 5 as a significant factor associated with worse PFS and OS [29] but more studies are needed to confirm this data.

Derived neutrophils/(leukocytes minus neutrophils) ratio (dNLR) may be more relevant than NLR because it includes monocytes and other granulocytes subpopulations. High dNLR has been associated with worse survival in pancreas cancer, renal carcinoma, and melanoma [30]. Mezquita et al. published the first study to explore this parameter in NSCLC. In this retrospective study with 305 NSCLC patients treated with ICIs, a lung immune prognostic index (LIPI) was developed. The index is based on dNLR greater than 3 and LDH greater than the upper limit of normal (ULN), characterizing three groups (good, 0 factors; intermediate, 1 factor; poor, 2 factors). Median OS for poor, intermediate, and good LIPI was 3 months (95% CI, 1 month to not reached [NR]), 10 months (95% CI, 8 months to NR), and 34 months (95% CI, 17 months to NR), respectively, and median PFS was 2.0 (95% CI, 1.7–4.0), 3.7 (95% CI, 3.0–4.8), and 6.3 (95% CI, 5.0–8.0) months (both *p* < 0.001). Disease control rate was also correlated with dNLR greater than 3 and LDH greater than ULN [31].

All these studies have some limitations such as their retrospective nature and there are potential confounders, not taken into consideration for the analysis, such as the concurrent use of medications that could alter the levels of the blood biomarkers.

Further prospective studies are needed in order to validate the use of peripheral blood biomarkers and be considered suitable to select patients that may benefit from immunotherapy in the clinical practice.

## 3. C-Reactive Protein (CRP)

It is known that systemic inflammation has a negative impact on cancer prognosis and recent investigations have hypothesized that blood-based markers of systemic inflammation would predict adverse outcomes in patients treated with immunotherapy.

CRP is an acute-phase protein that reflects tissue damage and is influenced by different factors such as interleukin 1 (IL-1) and the tumor necrosis factor (TNF) [32,33,34].

The first retrospective study evaluated the role of CRP in a series of 56 patients with different tumors (28.6% NSCLC) treated with anti PD-1 and anti PD-L1 treatment, and established the optimal cut-off value in 10 mg/mL. In this cohort, median PFS and OS were 4.0 and 17.0 months in patients with normal CRP levels in comparison to 2.0 and 10.0 months respectively in patients with high CRP levels [35].

The B-F1RST study, which included 152 patients treated with atezolizumab in the first-line setting, observed that a decrease in serum CRP levels over 6 weeks was associated with longer PFS and OS. They analyzed the CRP ratio between C3D1 and screening, showing worse outcomes in those patients with CRO ratio < 0.5 in comparison to those with ratio >0.5. The medium (m) PFS was 14.1 versus 4.6 months (HR 0.43 [90% CI: 0.24, 0.77]) and the median OS was NE versus 15.9 months (HR, 0.30 [90% CI: 0.13, 0.72]) [36].

## 4. Lactate Dehydrogenase (LDH)

LDH is a classic inflammatory marker that is released in blood by rapidly growing tumors, so high LDH blood levels are usually related to the tumor burden. In addition, increased LDH activity provokes lactic acid production and acidification of the tumoral microenvironment, which can be associated with the activation of macrophage-mediated angiogenesis and the development of metastases [37,38,39].

Increased LDH production can be a direct marker of intratumoral hypoxia, which has been linked to an increased risk of resistance to radiotherapy and chemotherapy treatments. Moreover, low pH-induced by LDH activity protects mitochondria from oxidative stress and is also associated with resistance to hypoxia-induced apoptosis. The activation of macrophage-mediated angiogenesis is also related to a high tumoral burden and accelerated tumor growth [40,41].

Elevated pretreatment level of LDH has been associated with poor outcome in several cancer types and it has been considered an adverse prognostic factor, related to the extension of the tumor. Dong Soo L. et al. evaluated the correlation between LDH and tumor characteristics in 394 patients with advanced NSCLC. The baseline serum LDH levels showed no significant associations with age, gender, histology, tumor differentiation, or smoking history but they have found a correlation with the disease extension at diagnosis. The extension was determined by 18(F)-fludeoxyglucose positron emission tomography scans and classified from 1 to 7 based on the sum of metastatic region [42].

Zhang et al., in a meta-analysis from six studies with 1136 patients with advanced lung cancer treated with ICIs have demonstrated that an elevated pretreatment LDH level is correlated with significant shorter PFS (HR = 1.53, 95% CI 1.27–1.83, *p* < 0.001) and OS (HR = 2.11, 95% CI 1.43–3.11, *p* < 0.001) [43].

Nevertheless, many issues still remain unsolved included LDH cut-off value, the optimal frequency of LDH measurement, or the potential value of changes related to treatment.

## 5. Nutritional Status Biomarkers

Recent scientific evidence suggest that nutrition, systemic inflammation, and tumoral immune microenvironment (TME) play a prognostic role in disease evolution. Alifano et al. [44] retrospectively evaluated the impact of nutritional status (based on prealbumin levels), systemic inflammation (measured by CRP), and TME (based on CD8 T lymphocyte and mature dendritic cells (mDC)) in 303 stage IV NSCLC patients. They concluded that a systemic inflammatory/poor nutritional status can affect the density of CD8 + T cells and mDC of the immune microenvironment. Based on these parameters they identified subgroups of patients with different long-term outcomes. Indeed, patients with undetectable CRP, high prealbumin level (0.285 mg/L), and high CD8 level (0.96/mm^2^) had a 5-year survival rate of 80% [60.9–91.1] as compared to 18% [7.9–35.6] in patients with an opposite pattern of values.

Several studies suggest that nutritional status can be a determinant of survival in lung cancer patients [45,46]. In particular, low pre-albumin levels have been associated with early recurrence and poor prognosis in resected NSCLC patients [47,48].

The prognostic nutritional index (PNI), which is based on serum albumin concentrations and total lymphocyte counts in the peripheral blood is an immune-nutritional marker whose predictive value is being studied. Shoji et al. analyzed the relation between pretreatment PNI value and treatment response in a retrospective study enrolling 102 NSCLC patients treated with ICIs. The best cut-off value considered was 45.5. In this analysis, the overall response rate (ORR) was significantly better in the high-PNI group compared with the low-PNI group (26% vs. 7.69%, *p* = 0.0131), as well as the disease control rate (DCR) (68% vs. 32%, *p* = 0.0002) [49].

Matsubara et al. also investigated the impact of immune-inflammation-nutritional parameters on the prognosis of 24 NSCLC patients treated with atezolizumab. They retrospectively analyzed the prognostic effect of neutrophil/lymphocyte ratio (NLR), platelet/lymphocyte ratio (PLR), and modified Glasgow prognostic score (mGPS). mGPS is based on C-reactive protein (CRP) and albumin levels, which represent not only inflammatory status but also nutritional status. Patients with elevated CRP (>10 mg/L) and hypoalbuminemia (<35 g/L) are given an mGPS score of 2. Patients with serum CRP < 10 mg/L with or without hypoalbuminemia receive a score of 0. Patients with only elevated CRP levels receive an mGPS score of 1. They analyzed the correlation between the score value and time to treatment failure (TTF) and OS in patients treated with atezolizumab. The results showed that the low PNI group had significantly shorter TTF and OS than the high PNI group [TTF HR = 5.41, *p* = 0.0044; OS HR = 7.28, *p* = 0.0283)], while the high NLR group had shorter TTF and OS than the low NLR group [TTF: HR = 2.45, *p* = 0.0616; OS: HR = 3.45, *p* = 0.0237]. Furthermore, patients with high mGPS experienced significantly shorter TTF and OS than those with low mGPS [TTF: HR = 4.07, *p* = 0.0043; OS: HR = 22.9, *p* < 0.0001)]. Despite limitations within this study (retrospective analysis and small population sample), results suggest that host immune-nutritional status can be relevant when using immunotherapy [50].

Liver plays an important role in proteins formation and in the regulation of innate and acquired immunity, so the dysfunction of this organ can lead to a less effective response to immunotherapy. The ALBI (Albumin-Bilirubin) score is calculated using the serum albumin and bilirubin levels and is divided into four grades. In a retrospective study with 140 NSCLC patients treated with ICIs, the subgroup with good hepatic reserve (ALBI grade 1–2a) had a significantly superior progression-free survival (PFS) and overall survival (OS) compared with the subgroup with poor hepatic reserve (ALBI grade 2b or 3) (PFS, 5.3 versus 2.5 months, *p* = 0.0019 and OS, 19.6 vs. 6.2 months, *p* = 0.0002) [51].

## 6. Soluble PD-1/PD-L1

The membrane-bound molecules programmed death 1 (PD-1) and its ligand PD-L1 (PD-1/PD-L1) belong to the immune checkpoint pathway. PD-1 pathway downregulates effector T cells in immune response, causing immune suppression. Recent studies have revealed that membrane-bound PD-1 and PD-L1 also have soluble forms, generated by proteolytic cleavage of the membrane-bound form and they may be measured by ELISA chemiluminescence test [52].

The expression of serum PD-L1 (sPD-L1) seems to be associated with a poor prognosis in solid cancers and increasing evidence suggests that sPD-L1 could be a dynamic biomarker in treatment response [53]. In a clinical trial in locally advanced or inoperable NSCLC patients treated with radiotherapy, sPD-L1 levels were measured showing that patients with lower baseline sPD-L1 levels had longer OS [54].

Costantini et al. investigated the prognostic role of sPD-L1 in NSCLC patients treated with nivolumab. They measured baseline sPD-L1 and after 2 weeks of treatment, showing higher PD-L1 level in non-responders patients compared to responders with a cutoff value of 33.97 pg/mL (94 sensitivity %, 56% specificity). The ORR in patients with low sPD-L1 concentration was 90% in comparison with 30% in those patients who had high sPD-L1 levels (*p* = 0.002) [55].

In another study, the role of baseline and dynamic evolution of sPD-1 and sPD-L1 was also evaluated. A composite criteria (sCombo) was defined by sPD-L1 and/or sPD-1 positivity. Baseline score positivity was associated with shorter PFS [78 days, 95% CI: 55–109 versus 658 days, 95% CI: 222–not reached; *p* = 0.0002] and OS (HR 3.99, 95% CI: 1.63–9.80; *p* = 0.003). In the multivariate analysis, score positivity remained significantly associated with shorter PFS (HR 2.66, 95% CI: 1.17–6.08; *p* = 0.02) [56].

Despite the evidence suggesting the potential biomarker role of sPD-1 and sPD-L1, currently, data of the predictive value of a dynamic sPD-L1 expression are scarce. Therefore, more patient involvement in clinical trials is needed to have stronger evidence in this area.

## 7. Blood Tumor Mutational Burden (bTMB)

The most studied biomarker after PD-L1 expression in tissue is tumor mutational burden (TMB) defined as the number of mutations per DNA megabases that can be considered a proxy for neoantigen burden. Although preliminary data of the predictive role of TMB in advanced NSCLC patients led to the assessment of TMB clinical utility in phase III clinical trials [57,58], current clinical evidence fails to show a survival benefit based on tissue TMB values [59]. A possible explanation would be that lung cancer is a heterogeneous disease, and one tissue sample does not represent the total number of mutations that a whole tumor may contain.

Blood-based tumor mutational burden (bTMB) estimated by a next-generation gene sequencing panel is another experimental serum-based maker of response to ICIs treatments and could provide a more complete picture of the mutational landscape of both primary tumor and metastases [60]. Gandara and colleagues published the first retrospective study that analyzed the potential role of bTMB. Overall, 1000 plasma samples of patients involved in two studies evaluating the efficacy of atezolizumab versus docetaxel in a second-line setting (POPLAR and OAK trials) were analyzed. Those patients with a bTMB ≥16 mutations/megabase showed superior PFS (HR 0.64, 95% IC (0.46–0.91)) and OS (HR0.64 (0.44–0.93). In that analysis, there was no correlation between bTMB and PD-L1 expression [61].

The MYSTIC trial assessed the efficacy of a durvalumab plus tremelimumab combination or durvalumab monotherapy compared to chemotherapy in advanced NSCLC patients with PD-L1 positive tumors (PDL-1 ≥ 25%). Although this study did not reach primary endpoints of PFS and OS, in a retrospective analysis they showed that patients with high bTMB (≥16 mut/Mb or ≥20 mut/Mb) had better survival outcomes with the combination of durvalumab plus tremelimumab in comparison to chemotherapy [62].

B-F1RST trial was the first prospective trial evaluating bTMB as a biomarker predicting benefit of first-line atezolizumab in advanced NSCLC. Final analysis confirmed that patients with high bTMB (≥16 mut/Mb) had longer PFS (5.0 versus 3.5 months, HR 0.80, 90% CI: 0.54–1.18) and OS (23.9 versus 13.4 months, HR 0.66, 90% CI: 0.40–1.10) compared to patients with low bTMB (<16 mut/Mb) [36].

However, the NEPTUNE trial, comparing durvalumab and tremelimumab versus platinum-based chemotherapy in the first-line treatment, did not find benefit in OS with the use of ICIs in patients with high bTMB (≥20 mut/Mb) [63].

Based on these findings, the value of bTMB remains controversial. In addition, there are issues that remain to be solved such as the optimal cut-off, the potential correlation between tissue TMB and bTMB or quality control. Finally, it is important to remind that the tumor must shed DNA into the blood and therefore, the likelihood of detecting ctDNA is also dependent on the overall tumor burden.

## 8. IDO

Indoleamine 2,3-dioxygenase (IDO) has been proposed as a possible marker of activation of immunological pathways induced by IFN gamma, representing an immune escape mechanism of tumor cells, and a negative prognostic factor in advanced stage lung cancer [64,65,66]. IDO is an enzyme that catalyzes the first step in the tryptophan–kynurenine catabolism pathway, creating an immunosuppressant environment as it induces suppression of effector T cells and activates immunosuppressive cells (such as regulatory T cells) in the tumor microenvironment (TME). The activity of IDO is expressed by the kynurenine (kyn)/tryptophan (trp) ratio, which is measured by high-performance liquid chromatography tandem mass spectrometry (HPLC) [67,68]. The coexpression of IDO1 and PD-L1 has been associated with poor prognosis in advanced NSCLC patients [69].

The increment of trp catabolism derives in higher kyn serum concentration, being related to more advanced stage at diagnosis, poorer prognosis, and resistance to antitumoral treatments. Furthermore, preclinical studies have shown that increased IDO activity is associated with resistance to treatment with ICIs [70]. Botticelli et al. analyzed the basal levels of serum trp, kyn and quinolinic acid in 27 advanced NSCLC patients prior to receiving second line therapy with nivolumab. The multivariant analysis showed that the kyn/trp ratio was associated with early progression (*p* = 0.01). Moreover, kyn/trp were associated with prolonged responses to nivolumab and PFS was significantly longer than in those patients with higher values of kyn/trp (median of PFS not reached in lower values arm versus 3 months in arm; HR: 0.2; 95% CI 0.06–0.62; *p* = 0.001). In addition, similar results were observed when patients were stratified by quinolinic acid values (median PFS not reached in patients with lower values versus median PFS 3 months in patients with higher values; HR: 0.3; CI 0.1–0.9; *p* = 0.018) [71].

IDO emerged as an attractive target to be explored in combination with ICIs in order to enhance the efficacy of immunotherapy. The phase 1b ECHO-110 trial evaluated the combination of epacadostat (IDO inhibitor) with atezolizumab in 29 advanced NSCLC patients previously treated. Only one patient achieved partial response, which was maintained for 8.3 months, and 27.5% of patients achieved stable disease [72]. A phase III study, the ECHO -301/KEYNOTE-252 trial evaluated epacadostat plus pembrolizumab 200 mg in patients with metastatic melanoma. Unfortunately, the study did not achieve the primary endpoint of improving PFS compared to pembrolizumab in monotherapy [73]. Several explanations have been proposed including the high rate of negative PD-L1 patients (41% in ECHO-110 study) who may have a lower benefit with immunotherapy as the only treatment. On the other hand, the selected epacadostat dose to be used in combination with pembrolizumab was chosen from preclinical trials, therefore the potential role of higher doses remains unclear.

Nowadays there are several trials exploring different IDO inhibitors in combination with immunotherapy and/or chemotherapy (NCT03348904: Nivolumab plus epacadostat plus chemotherapy versus chemotherapy ± nivolumab plus placebo; KEYNOTE-715-06/echo306-06: Pembrolizumab plus epacadostat plus chemotherapy versus pembrolizumab plus chemotherapy), whose results are largely awaited.

## 9. Cytokines

Various studies have reported that cytokine profiles in the peripheral blood can reflect the systemic immune conditions of patients, and several interleukins have been associated with the clinical outcomes [74].

C-X-C motif chemokine ligand 8 (CXCL8, also known as IL-8) is an angiogenic polypeptide expressed in multiple tumors. IL-8 regulates the chemotaxis of human neutrophils and affects the promotion of angiogenesis, tumor cells dedifferentiation, and tissue invasion and development of metastases [75,76,77,78].

The increased level of serum IL-8 has been considered a poor prognosis factor in NSCLC patients treated with ICIs. Schalper et al. published an analysis of baseline serum IL-8 levels in samples from 1344 patients treated with nivolumab monotherapy or the combination of nivolumab plus ipilimumab in several phase III clinical trials: CheckMate 067 in melanoma patients; CheckMate 017 and CheckMate 057 in advanced NSCLC patients and CheckMate 025 in renal cells cancer patients. High baseline serum IL-8 levels (defined by being above 23 pg/mL) were identified in 27.1–34.3% of patients and were associated with shorter overall survival across treatments and tumor types. No correlation was found between baseline serum IL-8 levels and PD-L1 expression. Accordingly, IL-8 has been proposed as a new therapeutic target. New molecules involving IL-8 or IL-8 receptors (for example, selective inhibitors of the IL-8 receptor CXCR2) are actually under development, being the combination with ICIs the most promising approach [79].

Serum IL-6 levels have been also studied and can be associated with tumor stage, size, metastasis, and survival in different cancer types. In advanced NSCLC patients treated with chemotherapy, elevated serum IL-6 levels have been reported as a prognostic factor for OS [80,81,82]. Kang et al. demonstrated that serum interleukin-6 level at baseline could also be a predictor marker of the efficacy of ICIs in NSCLC. Patients with low interleukin-6 level (< 13.1 pg/mL) at baseline presented significantly superior PFS (6.3 vs. 1.9 months; *p* < 0.001) and OS (not reached vs. 7.4 months; *p* < 0.001) than those with high interleukin-6 level [83].

Given the limited number of studies with small samples sizes, more research is needed to clarify the precise role of interleukins in response to immunotherapy in NSCLC.

## 10. Circulating Tumor Cell (CTC)

CTCs are tumor cells derived directly from the tumor into the bloodstream that can settle at a distant site [84,85,86,87,88]. Their presence have been reported as an independent adverse prognostic marker in several cancer types, including NSCLC [89].

The use of CTCs for serial evaluation of tumor evolution during the ICIs-treatment has been evaluated in different studies. Taminga et al. investigated the role of CTCs in 104 advanced NSCLC patients treated with ICIs. CTCs were measured in aliquots of 7.5 mL of blood, with the CellSearch^®^ Circulating Tumor Cell Kit, being detected in one-third of the patients. The presence of baseline CTC (>1) was correlated with worse PFS and OS (HR = 1.6, *p* = 0.05 and HR = 2.2, *p* < 0.01 respectively) as well as the increasement during the treatment (PFS HR = 3.4, *p* < 0.01 and OS HR = 3.7, *p* < 0.01 respectively) [90].

However, clinical applicability is limited due to different reasons including the low percentage of patients with baseline CTC found in 7.5 mL of blood (30% in this study). New techniques such as the diagnostic leukapheresis (DLA) could be an option. With this technique, mononuclear cells (MNCs) with a density of 1.005–1.08 g/mL are collected from peripheral blood via continuous centrifugation. As epithelial cells have similar density compared to MNCs, CTCs could be co-collected along with the targeted MNCs during the procedure [91].

Apart from the potential prognostic value of the number of CTC, it is well-known that PD-L1 can be assessed on 45–93% of CTCs samples. Some evidence suggests that high PD-L1 expression on CTCs at baseline is associated with a poor outcome in patients treated with nivolumab [92]. In addition, a dynamic increase in PD-L1+ CTCs during the treatment might indicate resistance to ICIs [93].

## 11. Exosomes

Exosomes are membrane-bound phospholipid vesicles secreted by most cell types, in particular tumor cells, and include proteins, nucleic acids (microRNA) and lipids. Tumor-derived exosomes play an important role in the communication between tumor cells and their microenvironment, favoring tumor progression. Moreover, they provided a protective vesicle for transporting small RNAs against degradation of RNAs and can be isolated in most biological fluids including serum and plasma by ultracentrifugation and commercial kits (for example, ExoQuick™ (System Biosciences, Palo Alto, CA, USA)). This makes exosomes an ideal specimen for liquid biopsy [94].

A study evaluated the role of exosomal microRNAs in 30 patients with advanced NSCLC who received PD-1/PD-L1 inhibitors. Plasma samples of these patients were collected before the administration of immunotherapy and every three cycles if the patients achieved partial response. Exosomes were prepared by ultracentrifugation, and exosome-derived miRNAs were profiled by RNA NGS. To identify the potential predictors of response, the patient samples were divided into PR (partial response) group and PD (progression disease) group. They identified three miRNAs from hsa-miR-320 family (hsa-miR-320d, hsa-miR-320c, and hsa-miR-320b) as potential predictors of response: all exhibited upregulation in PD group compared with PR group and were correlated with an unfavorable response to ICIs. Based on these findings, the authors suggested that patients with low level of hsa-miR-320d, hsa-miR-320c, hsa-miR-320b, and hsa-miR-125b-5p could be better candidates for immunotherapy [95].

Another study evaluated the PD-L1 mRNA expression in circulating exosomes in patients with melanoma (18 patients) or NSCLC (8 patients), treated with pembrolizumab and nivolumab. The data showed that, after treatment, PD-L1 mRNA expression in exosomes significantly decreased in responders, remained unchanged in those with stable disease, and significantly increased in patients with progressive disease [96]

A plasma immune-related microRNA-signature classifier (MSC) risk level and tumor PD-L1 expression were prospectively assessed in a consecutive series of 140 advanced NSCLC patients before starting treatment with ICIs. The results showed that patients with MSC intermediate/low risk level reported a significant reduction in disease progression and mortality compared to those with high MSC risk level. The corresponding multivariate HR were 0.35 (95% CI: 0.18–0.70; *p* = 0.0026) and 0.28 (95% CI: 0.12–0.58; *p* = 0.0007). Significant reduction in disease progression (multivariate HR = 0.35; 95% CI: 0.19–0.63; *p* = 0.0006) and mortality (multivariate HR = 0.43; 95% CI: 0.21–0.88; *p* = 0.0211) was also observed in patients with PD-L1 ≥ 50% compared to those with PD-L1 < 50%. When the two markers were considered together, patients with at least one favorable marker reported had a significant lower probability of disease progression (HR = 0.25; 95% CI: 0.12–0.56; *p* = 0.0006) and mortality (multivariate HR = 0.28; 95% CI: 0.12–0.65; *p* = 0.0034), compared to those with no favorable markers. They concluded that plasma MSC could complement PD-L1 expression to identify patients with no beneficial outcome from immunotherapy [97].

Shukuya et al. analyzed pretreatment plasma of 29 advanced NSCLC patients treated with single agent anti PD-1 or PD-L1 antibody. A total of 32 miRNAs (*p* = 0.0030–0.0495) from whole plasma and 7 extra vesicles-associated miRNAs (*p* = 0.041–0.0457) showed significant concentration differences between responders and non-responders and could have potential as predictive biomarkers for anti PD-1/PD-L1 treatment response [98].

Fan et al. analyzed 80 patients with stage IV NSCLC treated with nivolumab. Sera RNA was collected prior to the initiation and during the treatment. Responders had increased sera expression levels of miR-93, −138–5p, −200, −27a, −424, −34a, −28, −106b, −193a–3p, and -181a from pre-treatment to post-treatment (*p* < 0.01). Likewise, statistically significant improvement in PFS of patients was associated with the 10-high expressed miRNA pattern (median PFS of 6.25 versus 3.21 months, *p* < 0.001; hazard ratio, HR, 0.45; 95% CI, 0.25–0.76) and OS improvement was also significantly associated with the 10-high expressed miRNA pattern in responders versus non-responders (median OS of 7.65 versus 3.2 months, *p* < 0.001, HR, 0.39; 95% CI, 0.15–0.68) [99].

As conclusion, the detection of plasma-derived exosomal miRNAs can be a more accurate and dynamic reflection of the status of tumor cells, rather than tumor tissue and can monitor the tumor progression during the treatment.

## 12. Circulating Neutrophils and Myeloid-Derived Suppressive Cells (MDSC)

Other immunoregulatory cells that can be detected during a chronic inflammatory state are MDSC. These are a group of heterogenic cells derived from immature myeloid progenitors and play an immunosuppressive function, increasing T- reg cells and directly inhibiting the proliferation of T lymphocytes [100,101].

According to their morphological characteristics, human MDSC can be divided into neutrophil-like (g-MDSC or PMN-MDSC) and monocyte-like MDSC (M-MDSC).

The potential prognostic or predictive value of MDSCs and T—reg quantified by flow cytometry in peripheral blood has been also explored in NSCLC patients treated with ICIs [102].

Passaro et al. retrospectively analyzed the relationship between different populations of baseline white blood cells and granulocytic myeloid-derived suppressor cells (Gr-MDSC) in 53 stage IV NSCLC patients treated with nivolumab in a second-line setting. Outcome analysis showed that high baseline levels of Gr-MDSC and low baseline CD8/Gr-MDSC ratio are associated with significantly better (*p* = 0.02) response to immunotherapy treatment. Investigators observed a clear correlation between different immune markers and clinical outcome and described two prognostic GrMDSC-linked asset groups: A good prognostic group formed by patients with high baseline levels of Gr-MDSC (>/6 cell/uL), low absolute neutrophil count (<5840/uL), high eosinophil count (>90/uL) and NLR < 3; and a poor prognostic group which showed low baseline levels of Gr-MDSC (<6 cell/uL), absolute neutrophil count >5840/uL, eosinophil count <90/uL and NLR > 3. The multivariate analysis showed a statistically significant improvement for PFS (*p* = 0.003) and OS (*p* = 0.05) in favor of the identified good prognostic Gr-MDSC-linked asset group, compared with the poor prognosis group [103].

Another study, enrolling 63 NSCLC patients treated with nivolumab, analyzed the correlation between Lox-1+ PMN-MDSC and T—reg cells and the response to nivolumab.

Lectin-type oxidized LDL receptor-1 (Lox-1) is a specific marker expressed on immune-suppressive PMN-MDSCs that is not present in neutrophils; so, in order to isolate this type of cells, an anti-Lox1-PE mAb (Biolegend) is needed [104]. Before the treatment, the percentage of T—regs was higher in responders than in non-responder patients, but there was no significant difference in the frequency of Lox-1 +PMN-MDSCs. After the first dose of treatment, the median percentage of T—regs was also higher in responders, whereas the median percentage of Lox-1+ PMNMDSCs was significantly lower in responders than in non-responders. The ratio of T—regs to Lox- 1+ PMN-MDSCs (TMR) was also evaluated. The elevation of TMR ratio (cutoff 0.39) was associated with longer median PFS (103 versus 35 days; *p* = 0.0079) compared to patients with low TMR [105].

With this information, we can hypothesize that analyzing different cellular populations in peripheral blood could help us to classify patients into various subgroups depending on the expected benefit from immunotherapy.

## 13. Conclusions

Treatment involving the enhancement of the immune system against tumoral cells has revolutionized the treatment landscape of lung cancer, by achieving long-lasting responses and long-term survival in a subgroup of NSCLC patients which are not clearly defined yet.

Consequently, the characterization of new markers that can allow a better selection of patients who will benefit from immune-related treatments is mandatory nowadays, in order to avoid unnecessary toxicity and overwhelming expenditures for the healthcare systems.

The development of new biomarkers measured in blood samples can provide us with more information about the initial tumor complexity and the evolution of the mutational patterns during the course of the disease and may become an attractive strategy where we should put our efforts for the years to come. However, numerous questions have to be answered before a definitive transfer into clinical routine practice. This makes essential the design of prospective clinical trials with larger samples size to allow more robust evidence to guide us in the future.

## Figures and Tables

**Figure 1 jcm-11-03245-f001:**
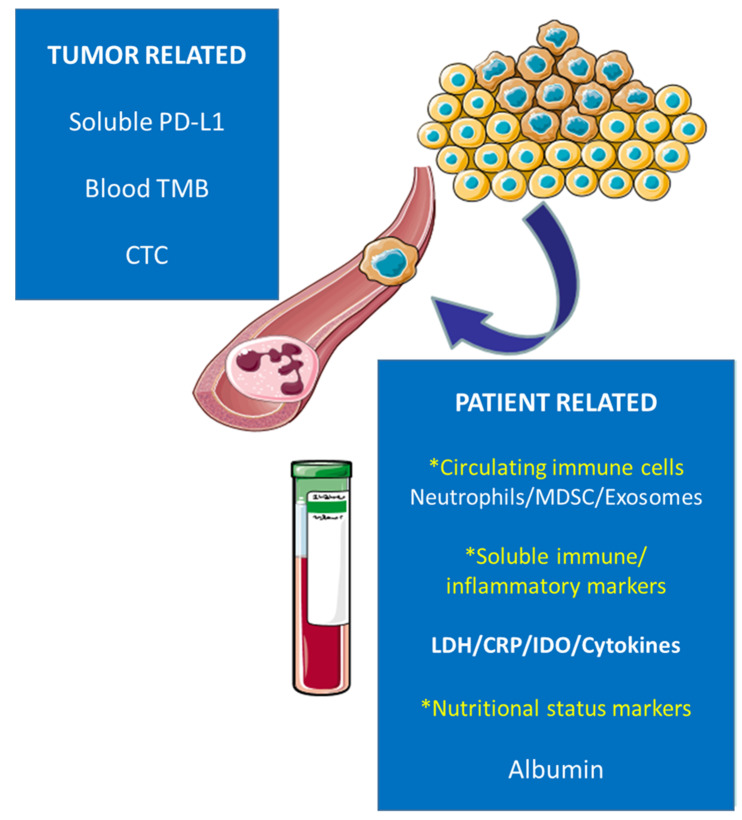
Potential blood biomarkers of clinical benefit in NSCLC patients treated with immunotherapy.

**Table 1 jcm-11-03245-t001:** Potential blood biomarkers of response to ICIs in NSCLC patients. TMB: tumor mutational burden. CTC: circulating tumor cells. MDSC: myeloid-derived suppressive cells. IDO: indoleamine 2,3-dioxygenase. LDH: lactate dehydrogenase. HPLC: high-performance liquid chromatography. NGS: next-generation sequencing. ELISA: enzyme-linked immunosorbent assay.

	Biomarkers		Method of Detection
		Soluble PD-L1	ELISA chemiluminescence
TUMOR-RELATED		Blood TMB	NGS
		CTC	Enrichment (CellSearch^®^) detection (IF staining)
	Circulating immune cells	Neutrophils	Flow cytometry
		MDSC	Ultracentrifugation
		Exosomes	ExoQuick™
PATIENT-RELATED	Soluble immune/inflammatory markers	LDH	Spectrophotometry
		CRP	Immunoturbidimetry
		IDO	HPLC
		CYTOKINES	ELISA chemiluminescence
	Nutritional status biomarkers	Albumin	Immunoturbidimetry

**Table 2 jcm-11-03245-t002:** Limitations of blood biomarkers clinical application. Level of evidence VI:Evidence from a single descriptive or qualities study.

Biomarkers	Limitations	Level of Evidence
SOLUBLE PD-L1	Data of predictive role of sPD-L1 are scarce Large-scale trials are needed	VI
BLOOD TMB	Dependent of overall tumor burden Optimal cut-off remains unsolved	VI
CTC	Low baseline CTC found in aliquots of 7.5 mL of blood New techniques of detection are needed	VI
NEUTROPHILS/MDSC	Prospective studies are needed	VI
EXOSOMES	Standard technologies must be established for the isolation/analysis of exosomes Mechanisms of exosomal miRNA delivery system remain incompletely understood Large-scale trials are needed	VI
LDH	Optimal cut-off value remain unsolved Large prospective studies are needed	VI
CRP	Establish optimal CRP ratio Large prospective studies are needed	VI
IDO	Big prospective studies are needed to confirm its role as predictive biomarker.	VI
CYTOKINES	Large-scale studies are needed	VI
ALBUMIN	Best cut-off value remain unsolved Large prospective studies are needed	VI

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
