# Peer review of "Blood Biomarkers of Response to Immune Checkpoint Inhibitors in Non-Small Cell Lung Cancer"

_jcm, 2022, doi:10.3390/jcm11113245_

Round 1
Reviewer 1 Report
The authors have properly addressed my concern and i would recommend for acceptance for publication.
Reviewer 2 Report
This a second submission after my previous comments.
Lage Alfranca et al. well improved the quality of this review according with previous remarks. I do not have any more comments .
This manuscript can now be considered for publication by editorial team.
Thanks for this review request
This manuscript is a resubmission of an earlier submission. The following is a list of the peer review reports and author responses from that submission.
Round 1
Reviewer 1 Report
The work by Alfranca et al. reviewed the “state of the art” of blood based biomarkers for predicting response to immunotherapy in NSCLC, an hot and rapidly evolving topic in the field. While the work is well written and sound interesting, it lacks a paragraph dedicated to the emerging role of circulating miRNAs as predictive biomarkers of ICI response. A bunch of papers have been recently published (some of them I reported in the list below), thus I strongly suggest to include a new paragraph describing this topic.
Boeri et al Clin Cancer Res 2019 10.1158/1078-0432.CCR-18-1981
Xiao-Xiao Peng et al J Immunother Cancer 2020 10.1136/jitc-2019-000376
Jinshuo Fan et al Genomics 2020 10.1016/j.ygeno.2019.11.019
Takehito Shukuya et al J Thorac Oncol 2020 10.1016/j.jtho.2020.05.022
Author Response
the role of microRNA as a biomarker of immunotherapy response is an interesting topic and i will include it in the manuscipt.
Reviewer 2 Report
Lage Alfranca et al proposed a review investigating the state of the art about blood markers helping to predict response with immunotherapies in NSCLC.
In general, this subject is timely and a hot topic. The blood / circulating aspects is original and needs to be detailed. However major limitations must be addressed:
- In general, a graphical abstract depicting all markers proposed in blood with a compartmentation between derived blood cells features, CTCs, derived tumor DNA/RNA product, soluble markers (cytokines…) might be of interest (these categories are a proposition and another one could be of interest)
- Keeping such an organization in the whole manuscript might allow to structure in comprehensives paragraphs helping to meet the readers ‘expectation. Currently, the review is only a succession of different ideas without link, several limited conclusions (for each paragraph) and a repeated sentence “larger prospective cohorts are needed”.
- For sections with many markers, comparative table should be included (name of the marker, technique of measure, context (neo-adj, adj, metastatic), association with ICI related responses, effectives of the trial…) adapted for each section.
- An update for many reference is needed (15, 38, 39, 66 to 70,
- Limitation paragraph should be added in every section of the MS.
- When authors exposed “ICI” from a specific trial, the specific drug should be mentioned in the whole MS
- In general, many point need to be updated:
- Introduction : ICI use in advanced stages and consolidation after radio chemotherapy are exposed. However, very actual and published data were reported in adjuvant context (Impower010) as also in Neoadjuvant condition (Checkmate 816).
- Introduction : response accordingtumor microenvironment is not evocated. Please add it. Other concept very timely in the ICI field are not exposed and must be considered such as T-cell exhaustion, the advantage of circulating marker to allow a monitoring inn a real-time manner.
- Many affirmatives sentence are not referenced. Please address it in the whole MS
- Authors have to provide numerical data, especially on main results with their confidence intervaland p-value when available.
- Which level for AMC
- “several studies have assessed the role of MDSCs…” : only one is discussed and reference… that is not convincing for a review
- Reference 16 : this reference do not support any data of “predictor factor of efficacy of ICIs” : Any patients of this study was exposed to ICI and all of them were in resectable (early) stages.. Please correct it
- Lox1+ and Lox-11 need to be defined and introduced
- I am not sure that the reference 18 was adapted to illustrate the feasibility and validity to measure sPD(L)1.
- The reference 20 and its related sentence are not based on patient treated by ICI. This seems out of concern for this review
- Inverse correlation between soluble and tissue PD-(L)1, with respective negative and positive prediction for ICI response in NSCLC could be mentioned and discussed with an hypothesize
- bTMB :
- Limitations about non-standardized techniques should be considered and discussed
- NEPTUNE trial : please include accurate data to illustrate the conclusion
- “evaluate” is underlined
- CRP section :
- “a serie of patient” : what are the effectives
- % of NSCLC must be added
- Which ICI ?
- Substantially : add p-value
- B-F1RST : authors also need to add data, p-value …
- LDH section :
- “baseline LDH level may predict the prognosis of patients treated with ICIs” : is this a hypothesize from the authors ?
- Reference 40 included only metastatic stage and can not support the extension at diagnosis. Such a correlation should be observed in comparison with early, advanced and metastatic stages.
- What is the reference supporting that ‘an early reduction of the basal levels in the first week after the start of the treatment has been related with clinical benefits” ?
- Nutritional status biomarkers :
- What is the relation with the “tumoral immune microenvironment”?
- The reference 46 is very interesting, focusing on immune nutrition and must be developed
- IDO section :
- Please develop the title section
- Are the HPL-CTMS an “easy-assessable and cost-effectiveness” technique able to be deployed in a daily practice ?
- CTC section :
- “their presence” : a cut-off was applied ? many parameters are available in CTCs and none of them are discussed. Please address it.
- Some paragraphs should be added in the final part of the MS :
- Current Clinical trial ongoing on this topic with their NCT ID.
- Markers associated with the prediction of toxicity could be mentioned and developed
- Authors should make a proposition for the most promising markers and/or how they should design trials / studies to screen and identify valuable markers in this competing field.
Thanks to the editorial team and the authors for the review request.